# Distillation Sparsity Training Algorithm for Accelerating Convolutional Neural Networks in Embedded Systems

Penghao Xiao , Teng Xu, Xiayang Xiao, Weisong Li and Haipeng Wang *

Key Laboratory for Information Science of Electromagnetic Waves (MoE), Fudan University,
Shanghai 200433, China; phxiao20@fudan.edu.cn (P.X.); xut22@m.fudan.edu.cn (T.X.);
xyxiao20@fudan.edu.cn (X.X.); weisongli20@fudan.edu.cn (W.L.)
* Correspondence: hpwang@fudan.edu.cn

**Abstract:** The rapid development of neural networks has come at the cost of increased computational complexity. Neural networks are both computationally intensive and memory intensive; as such, the minimal energy and computing power of satellites pose a challenge for automatic target recognition (ATR). Knowledge distillation (KD) can distill knowledge from a cumbersome teacher network to a lightweight student network, transferring the essential information learned by the teacher network. Thus, the concept of KD can be used to improve the accuracy of student networks. Even when learning from a teacher network, there is still redundancy in the student network. Traditional networks fix the structure before training, such that training does not improve the situation. This paper proposes a distillation sparsity training (DST) algorithm based on KD and network pruning to address the above limitations. We first improve the accuracy of the student network through KD, and then through network pruning, allowing the student network to learn which connections are essential. DST allows the teacher network to teach the pruned student network directly. The proposed algorithm was tested on the CIFAR-100, MSTAR, and FUSAR-Ship data sets, with a 50% sparsity setting. First, a new loss function for the teacher-pruned student was proposed, and the pruned student network showed a performance close to that of the teacher network. Second, a new sparsity model (uniformity half-pruning UHP) was designed to solve the problem that unstructured pruning does not facilitate the implementation of general-purpose hardware acceleration and storage. Compared with traditional unstructured pruning, UHP can double the speed of neural networks.

**Keywords:** neural networks; distillation sparsity training; uniformity half-pruning; general-purpose hardware acceleration



## 1. Introduction

Convolutional neural networks (CNNs) have achieved state-of-the-art results in a range of computer vision tasks, such as image classification [1,2], depth estimation [3,4], and object detection [5,6]. In the field of deep learning, the use of larger neural network models typically leads to higher accuracy in a variety of tasks [7–10]. Although the current CNNs have achieved remarkable results in the field of computer vision, these models depend on many parameters. Modern state-of-the-art models can comprise hundreds of billions of parameters, requiring trillions of computational operations per input sample. This limits their deployment in resource-limited devices and drives the need for model compression techniques. Model compression technology has also developed rapidly with the demand for intelligent terminals. In recent years, model compression has attracted significant research interest and many pertinent approaches have been proposed, such as KD, network pruning, and quantization [11,12].

KD aims to transfer knowledge from an influential teacher network to a smaller, faster student network, in order to expand its performance capability [13]. An obvious way to share the generalization capability of a complex model with a small model is to use the

probability of the complex model to generate a target as a "soft label" for training the small model. This soft target has high entropy, providing more information than the "hard label" used during training [14]. Early distillation methods aimed to transfer the last layer of the teacher [3]. However, the lack of oversight by intermediate layers hinders how the information of intermediate layers flows through the student network, reducing the learning potential of the student.

Although KD techniques can be used to transfer features from a complex network of teachers to a smaller network, the initial perception is that a robust teacher network with a high accuracy rate may provide better distillation results. However, when the student network does not have sufficient capacity to learn, further measures can be taken; for example, the student network can be pruned, as a network regularization technique [15,16]. Such approaches can provide more transferable knowledge for student networks with limited capacity. Network pruning [17] is a model compression technique that effectively removes the weights or neurons of a network, while maintaining its accuracy. After the initial training phase of the network, connections with all weights below a threshold are removed. This pruning transforms the dense network layers into sparse layers. The first phase involves learning the network's topology; that is, learning which connections are meaningful and removing the unimportant ones. Then, the sparse network is retrained, such that the remaining links can compensate for the deleted contacts. The pruning and retraining phases can be carried out iteratively, to further reduce the complexity of the network. In effect, this training process learns the network connectivity in addition to the weights, similarly to the case in the mammalian brain [18], where synapses are created in the first few months of a child's development, followed by gradual pruning of little-used connections, falling to typical adult values.

Network pruning is achieved by zeroing specific parameters, to enable the model to achieve sparsity [19,20]. However, it is difficult for the existing pruning algorithms to guarantee both model accuracy and inference performance (speed). Fine-grained sparsity maintains accuracy but is not conducive to memory access and cannot take advantage of general-purpose hardware to accelerate the computation. Therefore, it does not outperform traditional dense models using processor architectures such as GPUs [21]. Coarse-grained sparsity makes better use of processor resources but does not guarantee model accuracy by shearing off convolutional kernels. Overall, fine-grained sparsity ensures accuracy but does not guarantee model speed or efficient storage, while coarse-grained sparsity provides model speed and efficient storage but does not guarantee accuracy. We combine the best of both and design a specific UHP method, to be applied to the proposed algorithm based on general-purpose hardware acceleration conditions.

In this paper, we combine the advantages of KD and model sparsity to design a DST algorithm for accelerating CNNs in embedded systems. KD improves the performance of the student networks, while network sparsity enables efficient model storage and acceleration. The contributions of this paper include the following:

1. We design a unified training framework based on KD and network sparsity for model compression techniques.
2. Combined with general-purpose hardware, a uniform compression format is designed, to implement this pruning matrix for efficient storage and memory indexing.
3. The sparse model can be used for any general-purpose hardware acceleration. We tested the DST algorithm on an embedded system, and the acceleration result was significant.
4. The proposed algorithm was tested on the CIFAR-100, MSTAR, and FUSAR-Ship data sets. Its performance was significant on both SAR and optical data sets.

The remainder of this paper is organized as follows: Section 2 briefly reviews the related literature and summarizes the applications of KD and network pruning for model compression. Section 3 describes the motivation for the research presented in this paper and the limitations of existing algorithms. Section 4 describes the proposed DST algorithm in detail. Section 5 presents the experimental results and performance evaluation, and ana-

lytically verifies the DST algorithm. Section 6 concludes the paper and provides an outlook on future spaceborne ATR.

## 2. Related Work

### 2.1. KD

KD was initially proposed by Geoffrey Hinton [14], in order to enable a smaller network to learn the correlations between classes from the output of a larger pretrained teacher model. This work was extended in [15], to teach students to use intermediate representations as knowledge. They achieved this by minimizing the L2 distance between the feature maps of the student and teacher. In [15], it was shown that, if the gap between students and teachers is too large, the student's performance decreases. They suggested using intermediate features to distill knowledge between teachers and students. Each model uses a different architecture and has a separate set of weights. Slimmable neural networks [22] can execute on different widths by uniformly compressing the model width through a joint training approach. The smaller models benefited from the shared weights and the implicit KD provided.

In [23], a method to provide additional supervision for students using the feature map of the teacher was proposed. This method transfers only its intermediate layers with the help of encoding feature maps. Similarly, probabilistic knowledge transfer (PKT) [24] transfers the feature map of the penultimate layer (i.e., the layer before the classification layer) by matching its probability distribution. However, sharing the knowledge of an intermediate layer does not capture the critical connections between the layers. Attention transfer (AT) [25] involves an attention mechanism that transfers all intermediate layer representations to address this problem. Similarly, the hierarchical self-supervised augmented knowledge distillation (HSAKD) method [25] employs classifiers at the top of all intermediate layers to supervise the KD procedure. In addition, reference [26] introduced contrast representation distillation (CRD), which uses contrast loss to distill the feature maps from the last convolutional layer. All previous approaches considered the encoding of feature maps to match the width of teachers and students, allowing for a traceable architecture between students and teachers. Therefore, reference [27] introduced a method to efficiently encode the extracted features before the KD; however, encoding is still required.

### 2.2. Pruning

Neural networks often need to be more balanced, and there may be significant redundancy in deep learning models [28]. Network pruning removes channels and the corresponding weights that contribute insignificantly to the network accuracy. GoogLeNet [29,30] reduces the number of parameters of a neural network by employing an global average pool, instead of a fully connected layer. Network pruning is used to reduce the network's complexity and over-fitting. An early approach to pruning was biased weight decay [31]. Optimal Brain Damage [17] and Optimal Brain Surgeon [32] prune networks to reduce the number of connections based on the Hessian of the loss function, and it has been suggested that such pruning is more accurate than magnitude-based pruning approaches such as weight decay. However, the second-order derivative requires additional computation. There are two main branches of pruning, based on the granularity of the pruning: (1) unstructured pruning, which prunes individual weights; and (2) structured pruning, which prunes neurons (in most cases, channels of convolutional neural networks).

#### 2.2.1. Unstructured Pruning

Unstructured pruning [17] uses a single weight as the basic unit to delete weights and connections in the neural network, while maintaining the number of neurons in the network. General unstructured pruning consists of three steps: (1) training a large network model; (2) removing unnecessary connections (synapses) and weights (neurons), according to custom rules; and (3) finally fine-tuning the entire sparse neural network for updates. The iterative magnitude pruning (IMP) technique [33], which iteratively applies magnitude-

based trimming and fine-tuning, results in a significant performance enhancement. Lottery ticket rewinding (LTR) is an iterative magnitude pruning method with entitled repeated rolls [20,34]. IMP with learning rate (LR) rewinding, which recounts the learning rate schedule, has recently been shown to yield better results with more extensive networks [35]. However, the unstructured pruned network structure (i.e., the number of channels per layer) remains the same. As such, it is not easy to accelerate an unstructured pruning network without dedicated hardware [36].

### 2.2.2. Structured Pruning

Structured pruning [37–40] takes filters (i.e., neurons in channel units) as the basic unit and removes some filters that do not contribute much. Such an approach provides a smaller network with a more efficient network structure. This method can use general-purpose hardware acceleration and does not require specialized hardware and libraries to be designed. Similarly to magnitude-based unstructured pruning, the most straightforward approach is based on weight pruning filters [38,41]. Another method is to add a regularizer that induces sparsity during training [42–44]. Liu et al. [2] and Ye et al. [45] proposed a structured pruning scheme with filter-based batch normalization (BN) scale factors. Zhuang et al. [46] incorporated a polarization regularizer with a BN scale factor into structured pruning. However, the pruned network had more weights (parameters) than the unstructured pruning, due to the limitations of the network structure [34].

### 2.2.3. KD and Pruning

Previous works [47–50] combined KD and pruning for model compression. However, these approaches coincide with ours, in terms of technology and methodology. In [47], specific layers of the model were pruned, and a separate KD was performed on them. In [48], the pruned model was applied to the distilled student network. In [49], the teacher model was pruned to have the same width as the student network, in order to facilitate the transfer of intermediate layers. In [50], the teacher networks were also pruned, and the authors demonstrated that the pruned teacher networks had a regularization effect.

## 3. Motivation

### 3.1. Redundancy in the Student Network

KD [14] transfers knowledge from a strong teacher network to a smaller student network. The student network is trained with soft targets and some intermediate features provided by the teacher network [25,51,52]. Section 2.2.3 investigates and introduces the combined compression method, where knowledge flows from the teacher network to the student network. Even if KD is used to obtain the student network, there is still much redundancy. We propose an algorithm that can improve the network inference speed and provide an efficient storage format, while also improving the accuracy of the network.

### 3.2. Acceleration and Storage of Unstructured Pruning

As shown in Figure 1, unstructured pruning can make the dense weight tensor sparse. While this reduces the model size and computational effort, it also reduces the computational regularity, making parallelization in hardware more complex. For example, modern hardware architectures have many parallel processors, such as GPUs, with multiple processing unit (PEs). Numerous CUDA cores (also called arithmetic logic units or ALUs) exist in each PE of the general-purpose hardware acceleration unit, which can support multiple computations such as float32, float16, and/or int8 computations. Each PE handles part of the workload in parallel, allowing for faster computation than with serial execution of the workload. A pruning network leads to the potential problem of an unbalanced distribution of nonzero weights when computed in parallel on multiple PEs, resulting in a situation where each PE performs a different number of multiplicative accumulation operations. Therefore, the PE with fewer computational tasks must wait until the PE with the most computational tasks has finished before proceeding to the following computation stage.

As such, unbalanced workloads on different PEs may result in a gap between the actual and peak performance of the hardware. Traditional unstructured pruning stores the zero elements (as in Figure 1a) and the masks of the pruning process, resulting in a sizable final model, in terms of storage. This is even more detrimental regarding the deployment and acceleration of embedded systems.

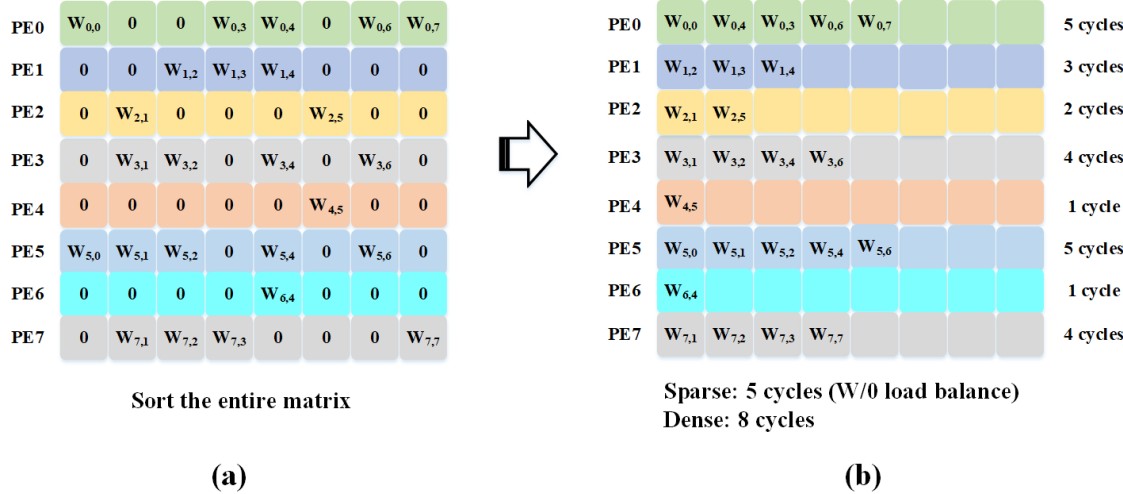

**Figure 1.** The sparse matrix of the pruning network is depicted in (**a**). Sorting operations performed on the whole matrix, such as that shown in (**b**) PE0–PE7, require at least five clocks for computation.

## 4. Methodology

### 4.1. Overview

In this paper, we design a unified end-to-end DST compression framework, as shown in Figure 2. We define a teacher network $f_t(\cdot; w_t)$ and a student network $f_s(\cdot; w_s)$. More formally, $f_t(\cdot; w_t)$ is a cumbersome network that needs to be compressed into a lightweight network $f_s(\cdot; w_s)$. Even after the KD and compression of the student network model $f_s(\cdot; w_s)$, much redundancy remains. This section provides a detailed architectural design based on the pruned student network $f_s(\cdot; w_p)$.

As shown in Figure 2, the design of the pruning algorithm for the student network is added to the KD (with respect to the student and teacher networks). The key idea of the student network pruning algorithm is to reconfigure the unstructured pruning architecture and construct a UHP method. After constructing the student network, each layer is pruned by 50% using the UHP method. Distillation learning is then performed on the pruned student network. Thus, the proposed compression algorithm includes three steps:

1. Train a teacher network and obtain $f_t(\cdot; w_t)$.
2. Apply the UHP algorithm to prune and obtain the pruned student network $f_s(\cdot; w_p)$.
3. Distill the teacher network $f_t(\cdot; w_t)$ to the student network $f_s(\cdot; w_p)$.

### 4.2. Distillation Formulation

As shown in Figure 2, this paper analyzes DST mathematically. Let $\{(x_i, y_i)\}_{i=1}^N$ be the data set, where the value of label $y_i$ comes from $\{1, 2, \ldots, K\}$. For each training instance $x$, the neural network $f(\cdot; w)$ outputs the probability of each label as $p(k \mid x) = \text{softmax}(z_k) = \frac{\exp(z_k)}{\sum\limits_{i=1}^{K} \exp(z_i)}$, where $z_i$ is the logit of the neural network $f(\cdot; w)$. Neural networks are trained by minimizing the cross-entropy loss $H(p_1, p_2) = -\sum\limits_{k=1}^{K} p_1[k] \log p_2[k]$. We are interested in a classification model with a $K$-dimensional probability distribution of the output. Let $f_{\text{true}}(x_i) \in \mathbb{R}^K$ be a one-hot encoding, where $f_{\text{true}}(x_i)[y_i] = 1$ denotes the real label $y_i$, and $f_{\text{true}}(x_i)[y'] = 0$ denotes all $y' \neq y_i$. Let $f_t(x; w)$ be the output of the teacher network.

When the input is $x$ and the weight is $w$, we train the teacher $f_t(\cdot; w)$ to achieve $w_t$ such that the cross-entropy loss is minimized.

When the input is $f_s(x; w_s)$ and the weight is $w_s$, $x$ is the output of the student network. For temperature $\tau$, the knowledge distillation loss is given by:

$$L_{KD}(w_s) = \frac{1}{N} \sum^{N} (1-\alpha) H(f_{\text{true}}(x_i), f_s(x_i; w_s)) + \alpha H(f_t(x; w_t), f_s(x; w_s)). \tag{1}$$

Further, through analysis of Equation (1):

$$
\begin{aligned}
L_{KD}(w_s) &= \frac{1}{N} \sum^{N} (1-\alpha) H(f_{\text{true}}(x_i), f_s(x_i; w_s)) + \alpha H(f_t(x; w_t), f_s(x; w_s)) \\
&= \frac{1}{N} \sum^{N} (1-\alpha) f_{\text{true}}(x_i) \log f_s(x_i; w_s) + \alpha f_t(x; w_t) \log f_s(x; w_s) \\
&= \frac{1}{N} \sum^{N} f_{\text{true}}(x_i) \log f_s(x_i; w_s) - \alpha f_{\text{true}}(x_i) \log f_s(x_i; w_s) + \alpha f_t(x; w_t) \log f_s(x; w_s) \\
&= \frac{1}{N} \sum^{N} ((1-\alpha) f_{\text{true}}(x_i) + \alpha f_t(x; w_t)) \log f_s(x_i; w_s).
\end{aligned}
\tag{2}
$$

When $f_\alpha(x; w_t) = (1-\alpha) f_{\text{true}}(x) + \alpha f_t(x; w_t)$, Equation (2) is equivalent to Equation (3):

$$L_{KD}(w_s) = \frac{1}{N} \sum_{i=1}^{N} H(f_\alpha(x_i; w_t), f_s(x_i; w_s)). \tag{3}$$

As shown by [53] and our derivation of Equations (1) and (3), in addition to teaching students knowledge, KD can be equated to label smoothing regularization (LSR). Thus, KD facilitates network label smoothing regularization training in students.

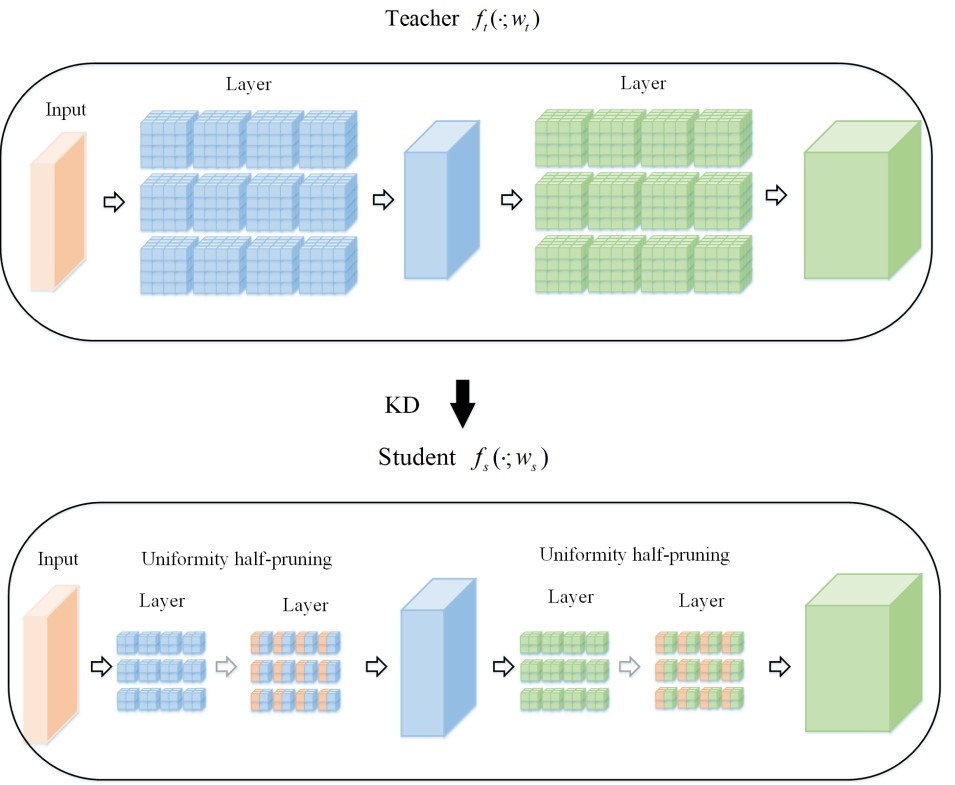

**Figure 2.** Overview of the DST strategy. The teacher network teaches the pruned student network, which can learn more from the teacher. The pruning algorithm used is the UHP method.

### 4.3. Pruned Student Distillation

This paper considers a pruned student network $f_s(x; w_p)$ to implement distillation training. The pruned student network $f_s(x; w_p)$ can also learn from the teacher network $f_t(x; w_t)$. We propose a new network compression framework for unstructured pruning of the student network $f_s(x; w_s)$. The critical challenge is obtaining the pruned student network $f_s(x; w_p)$ to learn knowledge from the teacher network $f_t(x; w_t)$. The pruned student network $f_s(x; w_p)$ is added to the loss function based on the above considerations. Thus, the distillation loss function is redesigned as follows:

$$L_{KD}(w_p) = \frac{1}{N} \sum_{i=1}^{N} H(f_\alpha(x_i; w_t), f_s(x_i; w_p)), \tag{4}$$

where $f_\alpha(x; w_t) = (1 - \alpha) f_{\text{true}}(x) + \alpha f_t(x; w_t)$.

### 4.4. Pruning Formulation

The pruned network is obtained using a $m \in \{0, 1\}^{|w|}$ binary mask applied to the student network's weight $w_s$. Although this mask can be applied to all weights in the network, we restrict our attention to the convolutional layers, as they contribute the most to the overall computational cost. Pruning aims to learn the weight $w_p$ that contributes most to the current objective of achieving a comparable performance to the original model. With the above considerations, we define Equation (5):

$$L(f_s(x, w_s \cdot m)) \approx L(f_s(x, w_p)), \frac{m_0}{|w|} = p, \tag{5}$$

where $p \in [0, 1]$ is a predefined pruning rate that controls the trade-off between the number of weights used, the computational complexity, and the expressiveness of the model.

### 4.5. Uniformity Half-Pruning

#### 4.5.1. Pruning Structure

The advantage of unstructured pruning is its high pruning ratio (e.g., 90%). Typically, 90% of unstructured sparsity is used for model inference. However, with general-purpose hardware acceleration architectures, such as GPUs and TPUs (tensor cores and systolic arrays, respectively), the gains in inference speed when using unstructured pruning may be more apparent. We start by initializing a random network and training the model with a fixed sparse pattern. The UHP sparse model architecture is obtained, to efficiently utilize hardware during training and inference. Based on the advantages of general-purpose hardware, the GPU kernel is optimized to accelerate this pruning model on the CUDA kernel. This solves the speed problem of unstructured pruning on general-purpose processors at the algorithmic level.

In this paper, we set the sparsity of only two nonzero weights among the four weights, to address the challenge posed by the unstructured pruning sparsity described in Section 3.1. The UHP pattern specifies that, for each group of 4 values, at least 2 must be 0. This results in a 50% sparsity, which makes it more practical to maintain accuracy without hyperparametric exploration than when using a higher sparsity. When accelerating a matrix, the UHP model has the following benefits over other sparsity methods: (1) efficient memory access; (2) low overhead compression format; and (3) $2\times$ higher computational throughput in a general-purpose processor architecture.

#### 4.5.2. Efficient Storage and Accelerating Computation

The UHP matrix (W) storage format is shown in Figure 3. In this mode, only 2 nonzero values from each group of 4 values need to be stored. The metadata in decoded compressed format are stored separately, using 2 bits to encode the location of each nonzero value in the 4 sets of values. For example, the metadata of the first row of the matrix in Figure 1 are

given as [[0; 3]; [0; 2]]. When performing matrix multiplication, metadata information is required to obtain the corresponding values from the second matrix.

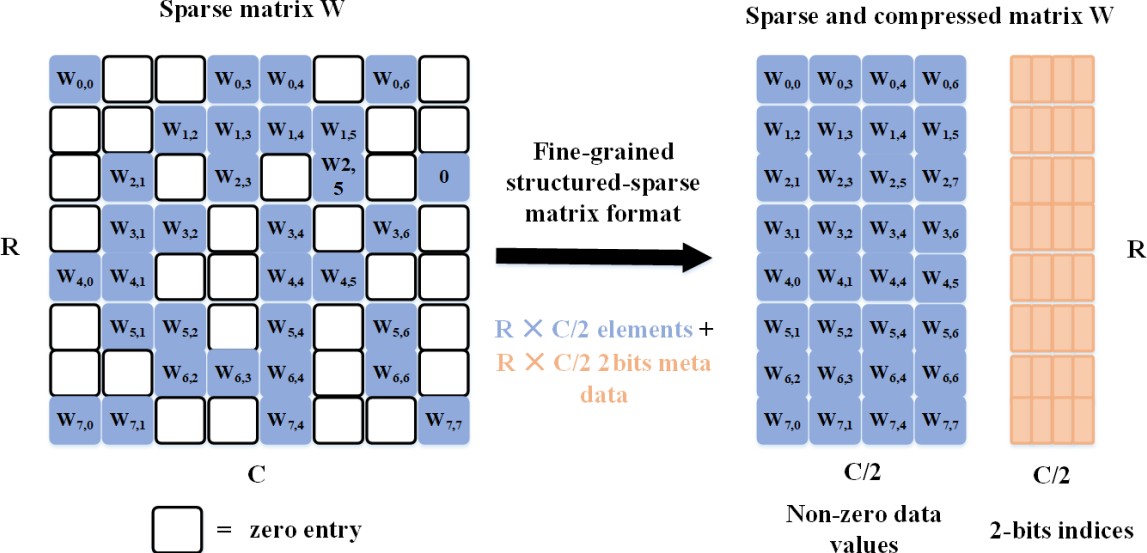

**Figure 3.** UHP matrix (W) storage format. The size of the uncompressed matrix is $R \times C$, and the size of the compressed matrix is $\frac{R \times C}{2}$.

The UHP matrix (W) storage format allows for efficient memory access. The unstructured sparse pattern results in low utilization of cache lines when accessing memory, therefore promoting low utilization of memory bandwidth. In addition, unstructured schemas usually use CSR/CSC/COO storage formats [54], which leads to data-dependent access and an increased latency of matrix reads. In contrast, each sub-block of the UHP matrix has the same sparsity level, allowing the hardware to take full advantage of large memory reads. Again, as the sparsity is constant throughout the matrix, the location of a nonzero value in the memory can be determined directly from the metadata.

The UHP matrix (W) allows for use of efficient storage formats. As the UHP matrix (W) storage format (shown in Figure 3) requires only 2 bits of metadata per value index; in the case of 32-bit operands, storing the sparse tensor in a compressed format results in a 47% saving in storage space: 4 Bytes require $4 \times 32$ bits = 128 bits of storage space, while the sparse UHP matrix (W) results in $2 \times 32$ bits + $2 \times 2$ bits = 68 bits to store the 2 nonzero weights.

### 4.5.3. UHP Computation Overhead

In this paper, we designed a convolution operation for the program, as shown in Figure 4. During model inference, the program loads the matrix, indexes the nonzero elements for matrix multiplication operations, and performs accumulation operations. The program unifies the design instructions. These instructions are the basis for the neural network layers involving mathematical operations, mainly matrix convolution operations. A network with 50% sparsity can halve the number of multiplication operations, thus doubling the computational performance of the hardware in the same network.

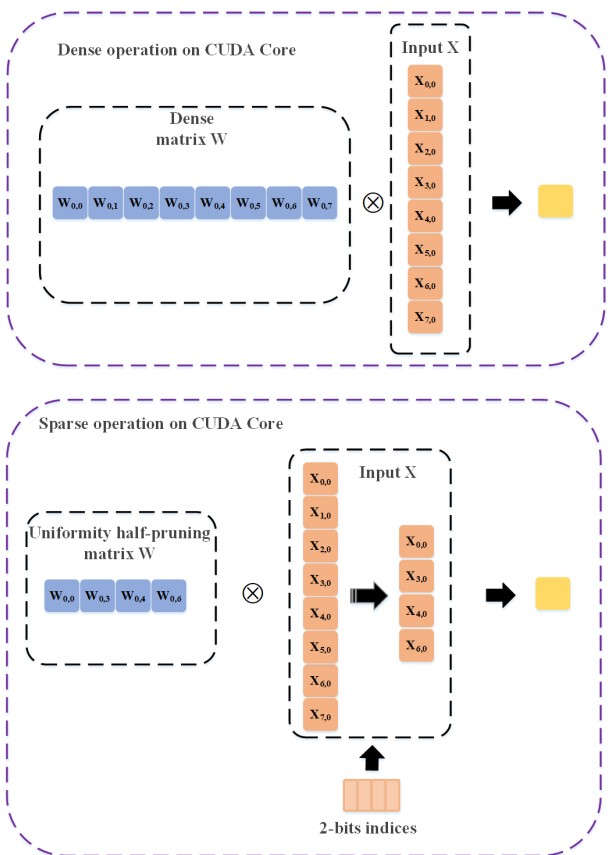

**Figure 4.** General-purpose hardware acceleration. The algorithm selects elements from the matrix X corresponding to nonzero values of the matrix W for the convolution operation by indexing, skipping unnecessary zero multiplication operations.

### 4.6. DST Workflow

Figure 5 shows the DST training process. In this method, the network is pruned with a 50% sparse pattern, maintaining the original accuracy and avoiding the need to conduct a hyperparameter search. As the aim is to reduce the size of the neural network and the running time at deployment, we trade higher training costs for smaller models. First, we train a teacher network $f_t(\cdot; w_t)$, which has better performance and generalization ability. Second, the teacher network $f_t(\cdot; w_t)$ and student network $f_s(\cdot; w_s)$ are trained simultaneously (the student network $f_s(\cdot; w_s)$ was not pretrained) and distilled for training. When the student network $f_s(\cdot; w_s)$ reaches a certain accuracy, the student network $f_s(\cdot; w_s)$ is uniformly pruned by half to obtain $f_s(\cdot; w_p)$. Finally, fine-tuning and distillation training are performed on the pruned student network $f_s(\cdot; w_p)$. The algorithm is described in Algorithm 1.

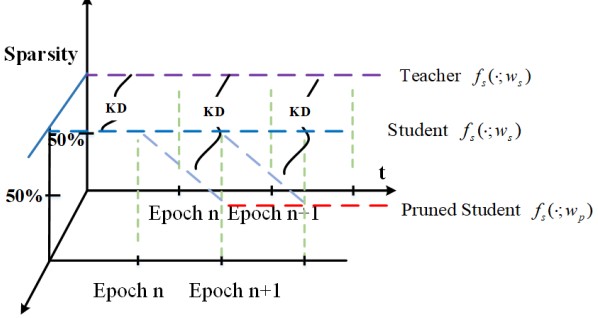

**Figure 5.** Timing of DST. Purple represents the teacher network $f_t(\cdot; w_t)$. Blue represents the student network $f_s(\cdot; w_s)$. Red represents the student network after pruning $f_s(\cdot; w_p)$.

---

**Algorithm 1:** Distillation Sparsity Training (DST)

---

**input**  : teacher network pretrained $f_t(x; w_t)$
           student network $f_s(x; w)$

**Initialize** : $f_t(x; w_t)$ KD $f_s(x; w)$, $L_{KD}(w_s) = \frac{1}{N} \sum_{i=1}^{N} H(f_\alpha(x_i; w_t), f_s(x_i; w_s))$

**if** $f_s(x; w_s) \leftarrow f_s(x; w)$ **then**
     $f_s(x; w) \leftarrow f_s(x; w_s)$ by UHP
**end**

**Initialize** : $f_t(x; w_t)$ KD $f_s(x; w)$, $L_{KD}(w_p) = \frac{1}{N} \sum_{i=1}^{N} H(f_\alpha(x_i; w_t), f_s(x_i; w_p))$

**if** $f_s(x; w_p) \leftarrow f_s(x; w)$ **then**
     **output:** $f_s(x; w_p)$
**end**

---

## 5. Experiment

### 5.1. Data Sets

The algorithm proposed in this study was evaluated on both the optical CIFAR-100 data set and the SAR MSTAR and FUSAR-ship data sets, in order to demonstrate the effectiveness of the DST algorithm. Experimental results were obtained to validate the proposed algorithm.

CIFAR-100 [55] consists of labeled subsets of the 80 million tiny image data set, collected by Alex Krizhevsky, Vinod Nair, and Geoffrey Hinton. This data set includes 100 categories, with each category containing 600 images. There are 500 training images and 100 testing images per class. The 100 classes in the CIFAR-100 are grouped into 20 super-classes. Each image comes with a "fine" label (the class to which it belongs) and a "coarse" label (the super-class to which it belongs).

The MSTAR (Moving and Stationary Target Acquisition and Recognition) data set [56] was collected and published by Sandia National Laboratory in the USA. The MSTAR data set can be divided into standard operating conditions (SOCs) and extended operating conditions (EOCs) data sets. There are 10 different ground military targets in the MSTAR data set, including BMP2 (tank), BTR70 (armored vehicle), T72 (tank), BTR60 (armored vehicle), 2S1 (artillery), BRDM (truck), D7 (bulldozer), T62 (tank), ZIL131 (truck), and ZSU234 (artillery). In the SOC, the training set has 2747 samples with a depression angle of $17°$, and the test set has 2425 samples with a depression angle of $15°$.

The FUSAR-Ship data set [57] was constructed and published by the Key Laboratory for Information Science of Electromagnetic Waves (MoE) of Fudan University, Shanghai, China. The FUSAR-Ship data set includes eight different ship targets: Bulk General, General Cargo, Container, Other Cargo, False Alarm, Fishing, Other Ship, and Tanker. We randomly selected 75% as the training set and 25% as the test set, such that the ratio of the training set to the test set was 3:1.

### 5.2. Experiment Setup and Implementation Details

For this experiment, we used a Jetson AGX Orin embedded device for evaluation of the proposed algorithm. The Jetson AGX Orin is based on an embedded platform with a 12-core ARM® v8.2 64-bit CPU, 32 GB of 256-bit LPDDR5 memory, and a 64 GB eMMC flash device running Ubuntu 20.04. In the evaluation, the inference processes were accelerated using CUDA and Tensor units for computation. We selected different teacher networks for different data sets, to achieve smooth knowledge transfer (e.g., ResNet18, ResNet34, ResNet50). The student networks were MobileNet, MobileNetV2, and MobileNetV3. One teacher network corresponded to three student networks, in order to test the effect of DST separately. For the CIFAR-100 data set, the momentum was 0.9 and the learning rate was 0.1 using the SGD optimizer, where learning rate was decayed by a factor of 5 at the 60th, 120th, and 160th epochs. The network was trained with an input size of $32 \times 32$,

batch size equal to 128, and weight decay of 0.0005 for 200 epochs. For the MSTAR and FUSAR-ship data sets, the momentum was 0.9 and the learning rate was 0.1 using the SGD optimizer, where the learning rate was decayed by a factor of 5 at the 60th, 120th, and 160th epochs. The network was trained with an input size of 224 × 224, batch size equal to 128, and weight decay of 0.0005 for 200 epochs. Unstructured pruning (50% pruning ratio) and UHP experiments were performed. To prevent overfitting of the network, we used the Pytorch library to implement dataset augmentation during training; for example, equal scaling of the images, random cropping, and random horizontal and vertical flipping.

### 5.3. Pruning Analysis

Figures 6–8 show the pruning ratio for the student networks MobileNet, MobileNetV2, and MobileNetV3, respectively, regarding the unstructured pruning (50% pruning ratio) and UHP for each layer. It can be seen that the pruning ratio of each layer under UHP was 50%. From the analysis in Section 4.5, our proposed algorithm is more suitable for general-purpose hardware acceleration. We set the overall pruning ratio for unstructured pruning to 50%; however, the pruning rate differed for each layer. According to the analysis in Section 3.1, the unstructured pruning algorithm does not accelerate well on general-purpose hardware. These results are validated in the experimental evaluation detailed in Section 5.4.

Figures 9–11 show views of the feature maps for the student networks MobileNet, MobileNetV2, and MobileNetV3, respectively, under unstructured pruning (50% pruning ratio) and UHP for each layer. Both sub-figures (a) and (b) show the same channel, where (a) is the channel with an unstructured pruning ratio higher than 50% and (b) is that with UHP. It can be seen that the features extracted under UHP are smoother and more focused than those with unstructured pruning, due to the random pruning per layer in the unstructured case. Figures 10 and 11 show that the relatively high ratio of unstructured pruning led to the direct loss of features in some layers, while the channel features under UHP remained relatively stable. This lack of concentration and the disappearance of features can lead to an inability to learn knowledge well in the process of DST. These results are validated in our analysis of the experimental evaluation in Section 5.5.

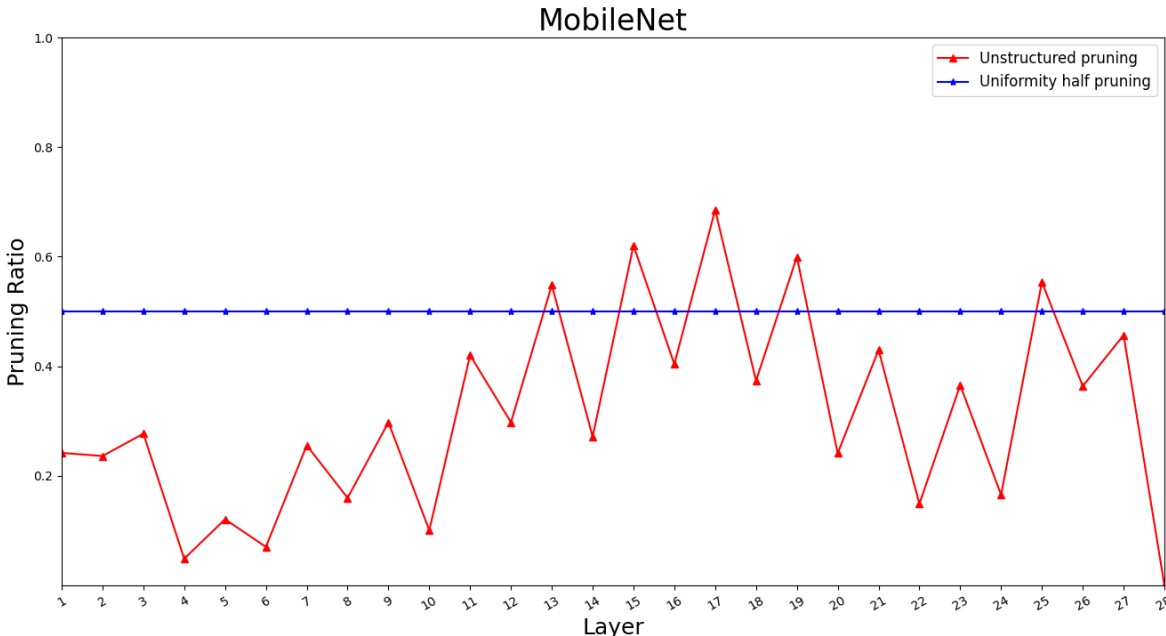

**Figure 6.** MobileNet network pruning ratio per layer. The blue line denotes UHP, and the red line denotes unstructured pruning.

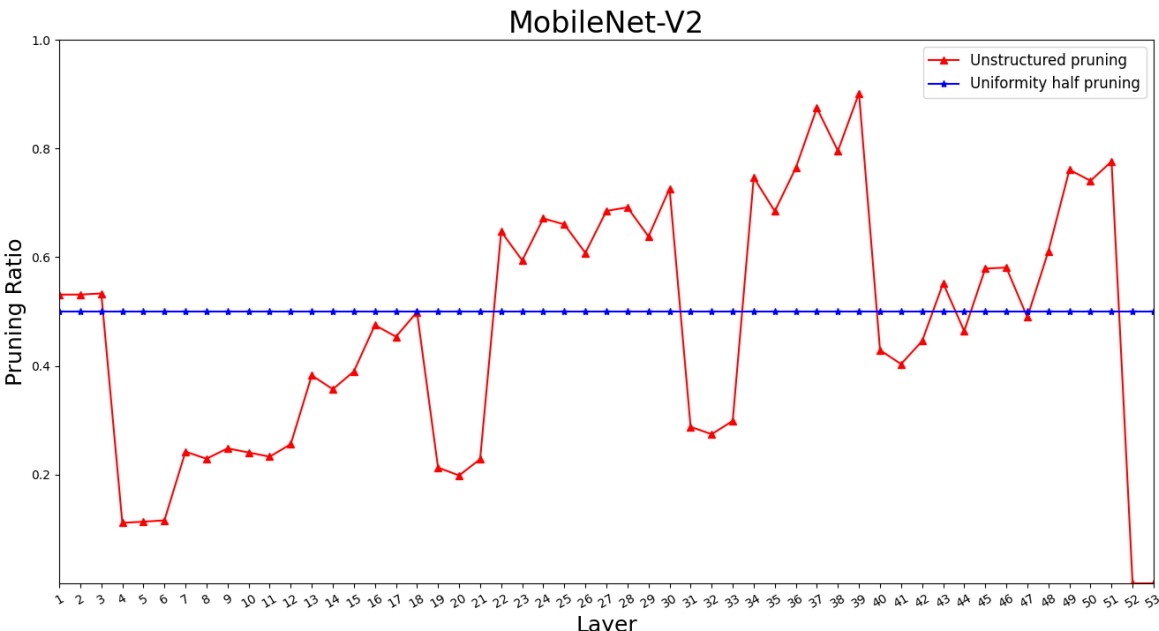

**Figure 7.** MobileNetV2 network pruning ratio per layer. The blue line denotes UHP, and the red line denotes unstructured pruning.

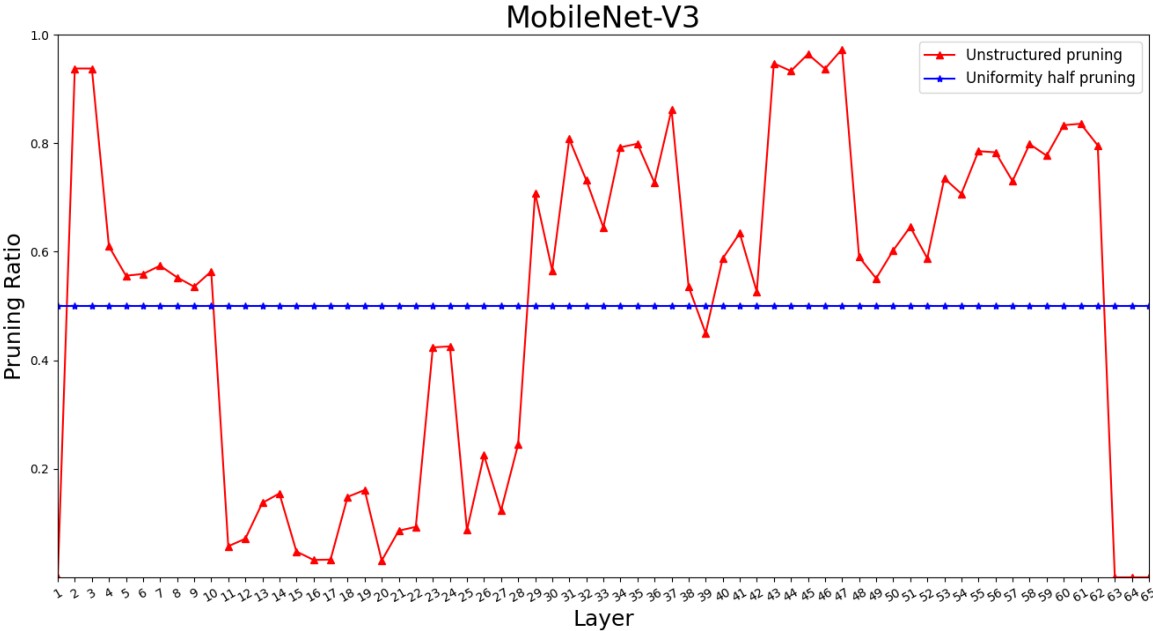

**Figure 8.** MobileNetV3 network pruning ratio per layer. The blue line denotes UHP, and the red line denotes unstructured pruning.

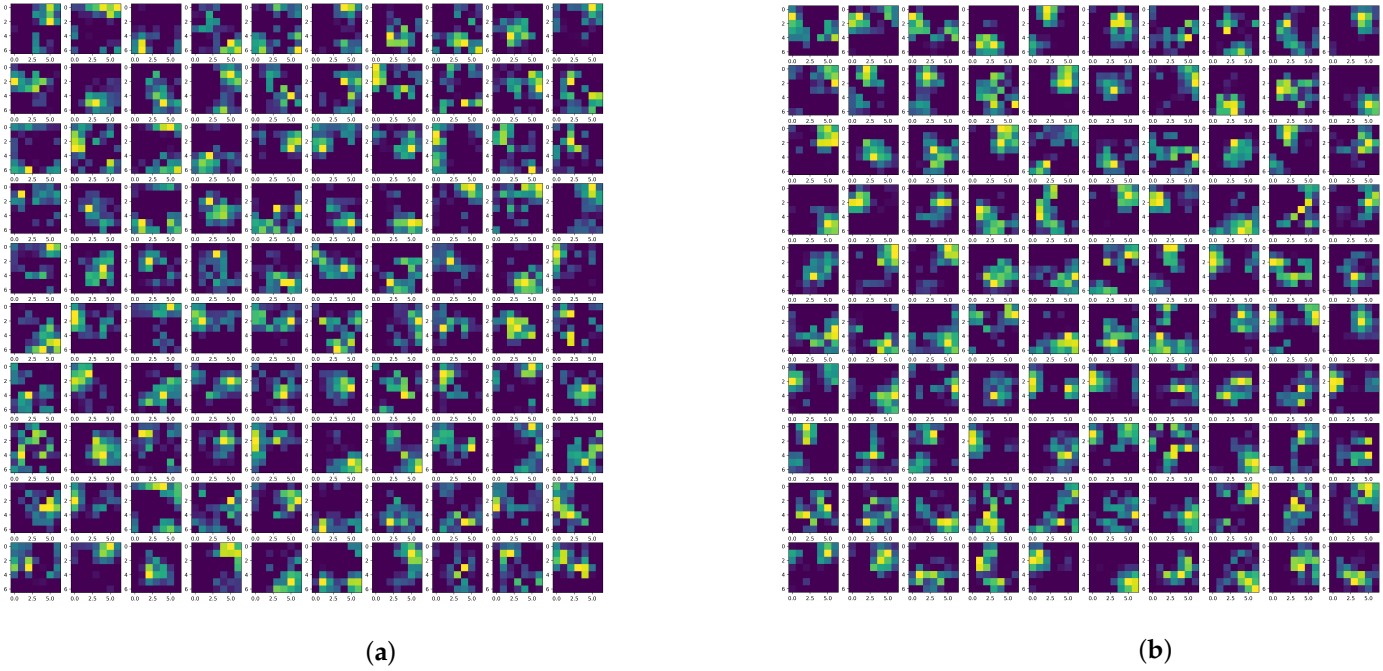

**Figure 9.** The feature maps of MobileNet after pruning: (**a**) Unstructured pruning; and (**b**) UHP.

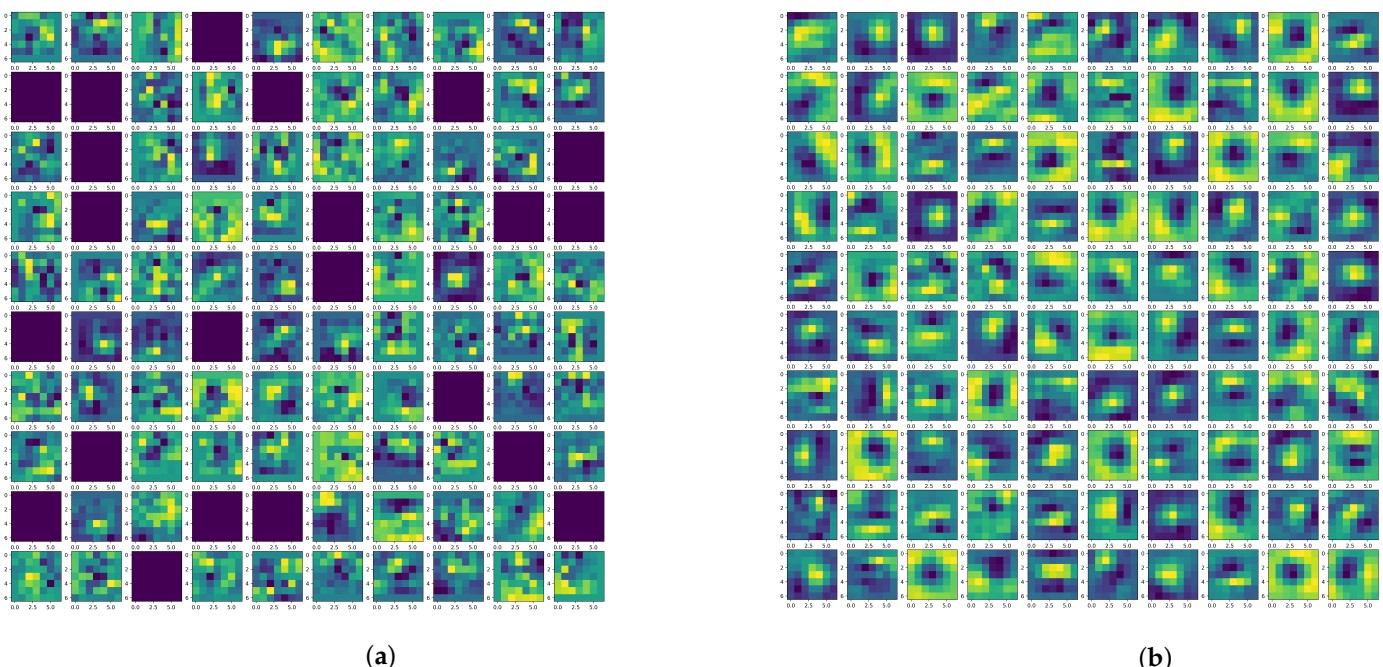

**Figure 10.** The feature maps of MobileNetV2 after pruning: (**a**) Unstructured pruning; and (**b**) UHP.

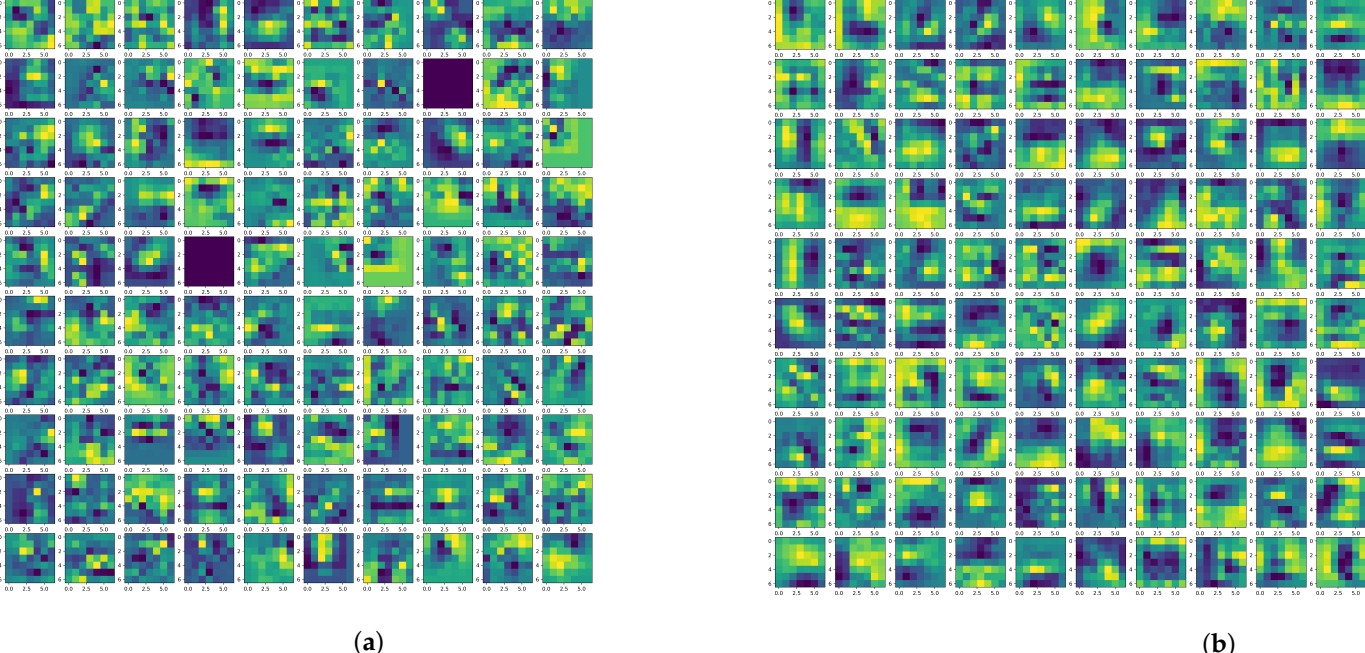

|         |         |
|:-------:|:-------:|
| (**a**) | (**b**) |

**Figure 11.** The feature maps of MobileNetV3 after pruning: (**a**) Unstructured pruning, and (**b**) UHP.

### 5.4. Computational Performance Evaluation

The network was deployed in an embedded system (Jetson AGX Orin), in order to validate the inference performance of UHP. In addition to evaluating the accuracy, we further evaluated the computational performance of the network. We validated the efficiency of our algorithm by calculating three metrics on the network. First, we assessed the model size by comparing the size of the pretrained, unstructured pruning, and UHP models. Second, we compared the number of network parameters for the pretrained, unstructured pruning, and UHP models. Third, we tested the frames per second (FPS), which is the computational efficiency of networks in embedded systems (Jetson AGX Orin), achieved by the three models with the embedded system.

Table 1 provides the model size, parameters, and FPS of the student networks. The MobileNet, MobileNetV2, and MobileNetV3 pretrained models were used as benchmarks. For the model size, unstructured pruning saved $-1.57\%$, $5.47\%$, and $2.99\%$ of the storage, while UHP saved $22.52\%$, $40.35\%$, and $37.72\%$ of storage, respectively. For the parameters, both were reduced by 50%. Regarding the FPS of the network, when the input of the network was $32 \times 32$, unstructured pruning led to an improvement of $6.03\%$, $2.67\%$, and $3.22\%$, while UHP led to an improvement of $130.00\%$, $110.70\%$, and $122.83\%$, respectively. When the network input was $224 \times 224$, unstructured pruning led to an improvement of $2.81\%$, $2.13\%$, and $5.49\%$, while UHP led to an improvement of $129.90\%$, $110.00\%$, and $111.37\%$, respectively. These experiments further validate the analysis provided in Section 5.3.

### 5.5. Accuracy Evaluation

#### 5.5.1. Results on CIFAR-100

We conducted nine groups of experiments on the CIFAR-100 data set. The teacher networks ResNet18, ResNet34, and ResNet50 were used to separately teach the student networks MobileNet, MobileNetV2, and MobileNetV3 by implementing DST. The set of experiments included assessment of the accuracy of single teacher and student networks. The accuracy of the networks was compared when implementing unstructured pruning (50% pruning rate) and UHP, when implementing DST, and when implementing unstructured pruning (50% pruning rate) and UHP and performing DST.

**Table 1.** Evaluation results of the computational performance for the student networks with the embedded system. Distillation sparsity training, DST; unstructured pruning (50% pruning ratio), UP; uniformity half-pruning, UHP. "✓" denotes baseline, and "-" denotes non-baseline.

| | | | MobileNet | MobileNetV2 | MobileNetV3 | Baseline |
|---|---|---|---|---|---|---|
| Model Size(MB) | | Pretrained | 12.70 | 9.69 | 16.70 | ✓ |
| | | DST-UP [48] | 12.90 | 9.16 | 16.20 | - |
| | | DST-UHP(Ours) | **9.84** | **5.78** | **10.40** | - |
| | | Relative | +1.57%/**−22.52%** | 5.47%/**−40.35%** | −2.99%/**−37.72%** | - |
| Parameters(M) | | Pretrained | 4.20 | 3.40 | 5.40 | ✓ |
| | | DST-UP [48] | 2.10 | 1.70 | 2.70 | - |
| | | DST-UHP(Ours) | **2.10** | **1.70** | **2.70** | - |
| | | Relative | −50.00%/**−50.00%** | −50.00%/**−50.00%** | −50.00%/**−50.00%** | - |
| FPS | Input 32 × 32 | Pretrained | 1160 | 675 | 622 | ✓ |
| | | DST-UP [48] | 1230 | 693 | 682 | - |
| | | DST-UHP(Ours) | **2668** | **1418** | **1386** | - |
| | | Relative | +6.03%/**+130.00%** | +2.67%/**+110.70%** | +3.22%/**+122.83%** | - |
| | Input 224 × 224 | Pretrained | 748 | 422 | 510 | ✓ |
| | | DST-UP [48] | 769 | 431 | 538 | - |
| | | DST-UHP(Ours) | **1720** | **886** | **1078** | - |
| | | Relative | +2.81%/**+129.90%** | +2.13%/**+110.00%** | +5.49%/**+111.37%** | - |

Table 2 shows the accuracy of the student network MobileNet under various conditions. Here, we used MobileNet without KD, UP, or UHP as a benchmark. After DST using the teacher networks ResNet18, ResNet34, and ResNet50, the accuracy increased by 4.00%, 3.21%, and 3.72%, respectively. Table 3 shows the accuracy of the student network MobileNetV2 under various conditions. Here, we took MobileNetV2 without KD, UP, or UHP as a benchmark. After DST using the teacher networks ResNet18, ResNet34, and ResNet50, the accuracy increased by 0.82%, 0.27%, and 0.55%, respectively. Table 4 shows the accuracy of the student network MobileNetV3 under various conditions. Here, we took MobileNetV3 without KD, UP, or UHP as a benchmark. After DST using the teacher networks ResNet18, ResNet34, and ResNet50, the accuracy increased by 1.58%, 1.68%, and 2.70%, respectively. These experiments validated the analysis in Section 5.3.

**Table 2.** Teacher networks ResNet18, ResNet34, ResNet50, and the student network MobileNet; all implemented the DST results. Unstructured pruning (50% pruning ratio), UP; uniformity half-pruning, UHP. "✓" denotes that the method is used, and "*" denotes that the method is not used.

| Teacher | Teacher Acc (Top-1) | Student | KD | UP [58] | UHP (Ours) | Student Acc (Top-1) | Relative |
|---|---|---|---|---|---|---|---|
| * | * | MobileNet | * | * | * | 67.58% [59] | Baseline |
| | | | * | ✓ | * | 68.26% [58] | +0.68% |
| | | | * | * | ✓ | 67.62% | −0.32% |
| ResNet18 | 76.41% | | ✓ | * | * | 71.62% [14] | +4.04% |
| | | | ✓ | ✓ | * | 70.75% [48] | +3.17% |
| | | | ✓ | * | ✓ | **71.48%** | **+4.00%** |
| ResNet34 | 78.05% | MobileNet | ✓ | * | * | 70.17% [14] | +2.59% |
| | | | ✓ | ✓ | * | 69.88% [48] | +2.30% |
| | | | ✓ | * | ✓ | **70.79%** | **+3.21%** |
| ResNet50 | 78.87% | | ✓ | * | * | 71.25% [14] | +3.67% |
| | | | ✓ | ✓ | * | 70.16% [48] | +2.58% |
| | | | ✓ | * | ✓ | **71.30%** | **+3.72%** |

**Table 3.** Teacher networks ResNet18, ResNet34, ResNet50, and the student network MobileNetV2; all implemented the DST results. Unstructured pruning (50% pruning ratio), UP; uniformity half-pruning, UHP. "✓" denotes that the method is used, and "*" denotes that the method is not used.

| Teacher | Teacher Acc (Top-1) | Student | KD | UP [58] | UHP (Ours) | Student Acc (Top-1) | Relative |
|---|---|---|---|---|---|---|---|
| * | * | MobileNetV2 | * | * | * | 68.90% [60] | Baseline |
|   |   |   | * | ✓ | * | 69.10% [58] | +0.20% |
|   |   |   | * | * | ✓ | 68.33% | −0.57% |
| ResNet18 | 76.41% |   | ✓ | * | * | 70.17% [14] | +1.27% |
|   |   |   | ✓ | ✓ | * | 68.93% [48] | +0.03% |
|   |   |   | ✓ | * | ✓ | **69.72%** | **+0.82%** |
| ResNet34 | 78.05% | MobileNetV2 | ✓ | * | * | 69.89% [14] | +0.99% |
|   |   |   | ✓ | ✓ | * | 68.00% [48] | −0.90% |
|   |   |   | ✓ | * | ✓ | **69.17%** | **+0.27%** |
| ResNet50 | 78.87% |   | ✓ | * | * | 69.75% [14] | +0.85% |
|   |   |   | ✓ | ✓ | * | 69.13% [48] | +0.23% |
|   |   |   | ✓ | * | ✓ | **69.45%** | **+0.55%** |

**Table 4.** Teacher networks ResNet18, ResNet34, ResNet50, and the student network MobileNetV3; all implemented the DST results. Unstructured pruning (50% pruning ratio), UP; uniformity half-pruning, UHP. "✓" denotes that the method is used, and "*" denotes that the method is not used.

| Teacher | Teacher Acc (Top-1) | Student | KD | UP [58] | UHP (Ours) | Student Acc (Top-1) | Relative |
|---|---|---|---|---|---|---|---|
| * | * | MobileNetV3 | * | * | * | 71.79% [61] | Baseline |
|   |   |   | * | ✓ | * | 72.03% [58] | +0.24% |
|   |   |   | * | * | ✓ | 71.75% | −0.04% |
| ResNet18 | 76.41% |   | ✓ | * | * | 73.41% [14] | +1.62% |
|   |   |   | ✓ | ✓ | * | 72.69% [48] | +0.90% |
|   |   |   | ✓ | * | ✓ | **73.37%** | **+1.58%** |
| ResNet34 | 78.05% | MobileNetV3 | ✓ | * | * | 73.54% [14] | +1.75% |
|   |   |   | ✓ | ✓ | * | 72.15% [48] | +0.36% |
|   |   |   | ✓ | * | ✓ | **73.47%** | **+1.68%** |
| ResNet50 | 78.87% |   | ✓ | * | * | 74.52% [14] | +2.73% |
|   |   |   | ✓ | ✓ | * | 73.96% [48] | +2.17% |
|   |   |   | ✓ | * | ✓ | **74.49%** | **+2.70%** |

### 5.5.2. MSTAR Results

Next, we implemented three groups of experiments using the MSTAR data set. The experimental process was similar to that in Section 5.5.1. As ResNet18 performed well on the MSTAR data set, the other teacher networks were not used. Table 5 shows the accuracy of the student network MobileNet under various conditions, taking MobileNet without KD, UP, or UHP as a benchmark. After DST, by implementing ResNet18 with the teacher network, the accuracy increased by 0.75%. Table 6 shows the accuracy of the student network MobileNetV2 under various conditions, taking MobileNetV2 without KD, UP, or UHP as a benchmark. After DST, by implementing ResNet18 with the teacher network, the accuracy increased by 2.85%. Table 7 shows the accuracy of the student network MobileNetV3 under various conditions, taking MobileNetV3 without KD, UP, or UHP as a benchmark. After DST, by implementing ResNet18 with the teacher network, the accuracy increased by 0.45%. These experiments validated the analysis provided in Section 5.3.

**Table 5.** Results for the teacher network ResNet18 and the student network MobileNet; both implemented the DST results. Unstructured pruning (50% pruning ratio), UP; uniformity half-pruning, UHP. "✓" denotes that the method is used, and "*" denotes that the method is not used.

| Teacher | Teacher Acc (Top-1) | Student | KD | UP [58] | UHP (Ours) | Student Acc (Top-1) | Relative |
|---------|---------------------|---------|-----|---------|------------|---------------------|----------|
| * | * | MobileNet | * | * | * | 98.56% [59] | Baseline |
|   |   |   | * | ✓ | * | 98.34% [58] | −0.22% |
|   |   |   | * | * | ✓ | 98.21% | −0.35% |
| ResNet18 | 99.78% | MobileNet | ✓ | * | * | 99.32% [14] | +0.76% |
|   |   |   | ✓ | ✓ | * | 99.24% [48] | +0.68% |
|   |   |   | ✓ | * | ✓ | **99.31%** | **+0.75%** |

**Table 6.** Results for the teacher network ResNet18 and the student network MobileNetV2; both implemented the DST results. Unstructured pruning (50% pruning ratio), UP; uniformity half-pruning, UHP. "✓" denotes that the method is used, and "*" denotes that the method is not used.

| Teacher | Teacher Acc (Top-1) | Student | KD | UP [58] | UHP (Ours) | Student Acc (Top-1) | Relative |
|---------|---------------------|---------|-----|---------|------------|---------------------|----------|
| * | * | MobileNetV2 | * | * | * | 96.54% [60] | Baseline |
|   |   |   | * | ✓ | * | 96.55% [58] | +0.01% |
|   |   |   | * | * | ✓ | 96.48% | −0.06% |
| ResNet18 | 99.78% | MobileNetV2 | ✓ | * | * | 99.40% [14] | +2.86% |
|   |   |   | ✓ | ✓ | * | 99.36% [48] | +2.82% |
|   |   |   | ✓ | * | ✓ | **99.39%** | **+2.85%** |

**Table 7.** Results for the teacher network ResNet18 and the student network MobileNetV3; both implemented the DST results. Unstructured pruning (50% pruning ratio), UP; uniformity half-pruning, UHP. "✓" denotes that the method is used, and "*" denotes that the method is not used.

| Teacher | Teacher Acc (Top-1) | Student | KD | UP [58] | UHP (Ours) | Student Acc (Top-1) | Relative |
|---------|---------------------|---------|-----|---------|------------|---------------------|----------|
| * | * | MobileNetV3 | * | * | * | 99.13% [61] | Baseline |
|   |   |   | * | ✓ | * | 99.06% [58] | +0.07% |
|   |   |   | * | * | ✓ | 99.05% | −0.08% |
| ResNet18 | 99.78% | MobileNetV3 | ✓ | * | * | 99.59% [14] | +0.46% |
|   |   |   | ✓ | ✓ | * | 99.55% [48] | +0.42% |
|   |   |   | ✓ | * | ✓ | **99.58%** | **+0.45%** |

### 5.5.3. FUSAR-Ship Results

We also conducted nine groups of experiments on the FUSAR-Ship data set. The experimental process was similar to that in Section 5.5.1. Table 8 shows the accuracy of the student network MobileNet under various conditions, taking MobileNet without KD, UP, or UHP as a benchmark. After DST using the teacher networks ResNet18, ResNet34, and ResNet50, the accuracy increased by 3.89%, 4.50%, and 4.20%, respectively. Table 9 shows the accuracy of the student network MobileNetV2 under various conditions, taking MobileNetV2 without KD, UP, or UHP as a benchmark. After DST using the teacher networks ResNet18, ResNet34, and ResNet50, the accuracy increased by 2.88%, 3.31%, and 4.20%, respectively. Table 10 shows the accuracy of the student network MobileNetV3 under various conditions, taking MobileNetV3 without KD, UP, or UHP as a benchmark. After DST using the teacher networks ResNet18, ResNet34, and ResNet50, the accuracy increased by 3.46%, 3.80%, and 4.52%, respectively. These experiments validated the analysis in Section 5.3.

**Table 8.** Results for the teacher networks ResNet18, ResNet34, ResNet50, and the student network MobileNet; all implemented the DST results. Unstructured pruning (50% pruning ratio), UP; uniformity half-pruning, UHP. "✓" denotes that the method is used, and "*" denotes that the method is not used.

| Teacher | Teacher Acc (Top-1) | Student | KD | UP [58] | UHP (Ours) | Student Acc (Top-1) | Relative |
|---|---|---|---|---|---|---|---|
| * | * | MobileNet | * | * | * | 70.42% [59] | Baseline |
| | | | * | ✓ | * | 70.35% [58] | −0.07% |
| | | | * | * | ✓ | 70.67% | +0.25% |
| ResNet18 | 74.38% | | ✓ | * | * | 74.82% [14] | +4.40% |
| | | | ✓ | ✓ | * | 73.59% [48] | +3.17% |
| | | | ✓ | * | ✓ | **74.31%** | **+3.89%** |
| ResNet34 | 75.31% | MobileNet | ✓ | * | * | 75.35% [14] | +4.93% |
| | | | ✓ | ✓ | * | 73.82% [48] | +3.40% |
| | | | ✓ | * | ✓ | **74.92%** | **+4.50%** |
| ResNet50 | 75.87% | | ✓ | * | * | 75.62% [14] | +5.20% |
| | | | ✓ | ✓ | * | 74.01% [48] | +3.59% |
| | | | ✓ | * | ✓ | **74.62%** | **+4.20%** |

**Table 9.** Results for the teacher networks ResNet18, ResNet34, ResNet50, and the student network MobileNetV2; all implemented the DST results. Unstructured pruning (50% pruning ratio), UP; uniformity half-pruning, UHP. "✓" denotes that the method is used, and "*" denotes that the method is not used.

| Teacher | Teacher Acc (Top-1) | Student | KD | UP [58] | UHP (Ours) | Student Acc (Top-1) | Relative |
|---|---|---|---|---|---|---|---|
| * | * | MobileNetV2 | * | * | * | 71.37% [60] | Baseline |
| | | | * | ✓ | * | 70.16% [58] | −1.21% |
| | | | * | * | ✓ | 69.27% | −2.10% |
| ResNet18 | 74.38% | | ✓ | * | * | 74.16% [14] | +2.79% |
| | | | ✓ | ✓ | * | 73.29% [48] | +1.92% |
| | | | ✓ | * | ✓ | **74.25%** | **+2.88%** |
| ResNet34 | 75.31% | MobileNetV2 | ✓ | * | * | 74.87% [14] | +3.50% |
| | | | ✓ | ✓ | * | 73.74% [48] | +2.37% |
| | | | ✓ | * | ✓ | **74.68%** | **+3.31%** |
| ResNet50 | 75.87% | | ✓ | * | * | 74.98% [14] | +3.61% |
| | | | ✓ | ✓ | * | 73.85% [48] | +2.48% |
| | | | ✓ | * | ✓ | **74.96%** | **+3.59%** |

According to the computational performance evaluation in Section 5.4 and the accuracy evaluation in Section 5.5, DST improved the accuracy of the pruned network. Regarding the model size of the student network, UHP saved close to 40% of storage space, compared to unstructured pruning. Considering the FPS of the network, UHP achieved a 2× increase in speed compared to the pretrained model, while the speed increase due to unstructured pruning was insignificant.

**Table 10.** Results for the teacher networks ResNet18, ResNet34, ResNet50, and the student network MobileNetV3; all implemented the DST results. Unstructured pruning (50% pruning ratio), UP; uniformity half-pruning, UHP. "✓" denotes that the method is used, and "*" denotes that the method is not used.

| Teacher | Teacher Acc (Top-1) | Student | KD | UP [58] | UHP (Ours) | Student Acc (Top-1) | Relative |
|---|---|---|---|---|---|---|---|
| * | * | MobileNetV3 | * | * | * | 71.12% [61] | Baseline |
| | | | * | ✓ | * | 70.19% [58] | −0.93% |
| | | | * | * | ✓ | 71.44% | +0.12% |
| ResNet18 | 74.38% | | ✓ | * | * | 74.57% [14] | +3.45% |
| | | | ✓ | ✓ | * | 73.18% [48] | +2.06% |
| | | | ✓ | * | ✓ | **74.58%** | **+3.46%** |
| ResNet34 | 75.31% | MobileNetV3 | ✓ | * | * | 74.98% [14] | +3.86% |
| | | | ✓ | ✓ | * | 74.15% [48] | +3.03% |
| | | | ✓ | * | ✓ | **74.92%** | **+3.80%** |
| ResNet50 | 75.87% | | ✓ | * | * | 75.92% [14] | +4.80% |
| | | | ✓ | ✓ | * | 74.13% [48] | +3.01% |
| | | | ✓ | * | ✓ | **75.64%** | **+4.52%** |

## 6. Conclusions

KD and network pruning are essential technical tools for model compression. In this work, we proposed a model compression algorithm that combines KD and network pruning, which we call DST. In particular, network pruning was applied to the student network to effectively transfer knowledge from the teacher network to the pruned student network and further compress the student network. We theoretically derived and designed a new teacher-pruned student objective function, in order to achieve compression while improving the accuracy of the student network. Considering that unstructured pruning cannot achieve general-purpose hardware acceleration, we reconstructed the unstructured pruning approach, to propose the UHP method. Extensive experiments on the CIFAR-100, MSTAR, and FUSAR-Ship data sets demonstrated that the proposed DST can achieve state-of-the-art compression performance in terms of prediction accuracy, inference acceleration, and storage efficiency. We also obtained the FPS values for the compressed models when using the proposed DST on embedded platforms, in order to demonstrate the real-time application potential of our compressed model on mobile devices.

Deep neural networks include CNNs, transformers, etc. The DST algorithm mainly implements model compression for generic CNNs, but the structure of the transformer differs significantly from CNNs, and the proposed algorithm requires further improvement in its compression structure. Compression techniques for general-purpose deep neural networks (e.g., CNNs, transformers) could be further investigated in the future. Currently, we have only performed tests on a GPU-based Jetson AGX Orin. Test studies have yet to be conducted for other embedded devices (e.g., FPGA, DSP). Spaceborne SAR technology may carry different embedded systems and use various deep neural networks, and we hope the proposed compression algorithm can achieve generality.

**Author Contributions:** Conceptualization, P.X. and H.W.; methodology, P.X.; software, P.X.; validation, P.X. and T.X.; formal analysis, P.X.; investigation, P.X. and T.X.; resources, P.X.; data curation, X.X. and W.L.; writing—original draft preparation, P.X.; writing—review and editing, P.X.; visualization, P.X. and T.X.; supervision, H.W.; project administration, H.W.; funding acquisition, H.W. All authors have read and agreed to the published version of the manuscript.

**Funding:** This work was supported in part by the National Natural Science Foundation of China (Grant No. 62271153) and the Natural Science Foundation of Shanghai (Grant No. 22ZR1406700).

**Data Availability Statement:** Please contact Haipeng Wang (hpwang@fudan.edu.cn) to access the data.

**Acknowledgments:** The authors would like to thank the anonymous reviewers.

**Conflicts of Interest:** The authors declare that there are no conflict of interests.

## Abbreviations

The following abbreviations are used in this manuscript:

| | |
|---|---|
| ATR | Automatic Target Recognition |
| KD | Knowledge Distillation |
| DST | Distillation Sparsity Training |
| CNNs | Convolutional Neural Networks |
| PKT | Probabilistic Knowledge Transfer |
| AT | Attention Transfer |
| HSAKD | Hierarchical Self-supervised Augmented Knowledge Distillation |
| IMP | Iterative Magnitude Pruning |
| LTR | Lottery Ticket Rewinding |
| LR | Learning Rate |
| BN | Batch Normalization |
| PEs | multiple Processing units |
| ALUs | Arithmetic Logic Units |
| FPS | Frames Per Second |
| UHP | Uniformity Half-Pruning |
| UP | Unstructured Pruning |

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
