# Peer review of "Distillation Sparsity Training Algorithm for Accelerating Convolutional Neural Networks in Embedded Systems"

_remotesensing, doi:10.3390/rs15102609_

Round 1

Reviewer 1 Report

In order to simplify the computational complexity of network, a sparse training algorithm based on knowledge distillation and network pruning is designed by the authors. Firstly, knowledge distillation is used to improve the accuracy of student network, and then "uniformity half-pruning(UHP)" is used to reduce the redundancy of student network. Finally, the effectiveness of the proposed algorithm is proved by experiments on three datasets. The content of this paper is sufficient and logical, but there were some problems to be solved:

1.     The authors should directly describe the specific contribution of this paper in the abstract;

2.     For the input of the three datasets, the size of the images in MSTAR dataset were 128*128 and the size of the images in FUSAR-ship data set were 512*512, but the authors described the size of the input images as 128*128. Did the authors process the dataset accordingly?

3.     The authors should make a brief summary and reduce the introduction of the article when introduced techniques of distillation learning and model pruning;

4.     The authors should give the full name of FPS(Frames Per Second) and properly explain the meaning of the evaluation metrics;

5.     In order to prevent the overfitting of the network, the dataset should be augmented;

6.     The main task of this paper was lightweight. The authors only chose the corresponding teacher and student models for experiments, but the authors should select some of the state-of-the-art networks to compare the model parameters.

 Minor editing of English language required

Reviewer 2 Report

The article discusses the rapid advancements in neural networks, highlighting their computational and memory-intensive nature. The authors propose a novel algorithm called Distillation Sparsity Training (DST), which combines knowledge distillation (KD) and network pruning to enhance the accuracy of student networks. The DST algorithm is evaluated on CIFAR-100, MSTAR, and FUSAR-Ship datasets with a 50% sparsity setting. The article concludes that the DST algorithm improves the performance of student networks, facilitating efficient model storage and acceleration.

The article primarily focuses on the implementation of Distillation Sparsity Training in neural networks, which may not directly relate to remote sensing. Generally, the article lacks empirical evidence demonstrating the extent of computational resource savings achieved by using DST compared to alternative methods. Additionally, it does not compare the algorithm's performance with other state-of-the-art methods in terms of accuracy or speed. Lastly, the article does not explore how DST can be applied to different types of neural networks or other applications.

Here are a few minor questions and suggestions for improvement:

1. The content does not sufficiently demonstrate its relevance to embedded systems research or application. Please address this issue.

2. In the introduction section, the authors could provide an overview of each section in the final paragraph, helping readers to follow the paper's structure more easily.

3. The conclusion is quite simplistic. The authors should offer more significant conclusions and highlight valuable suggestions for real-world engineering applications.

I hope these suggestions will aid in the revision of the manuscript.

Extensive editing of the English language is required.

Round 2

Reviewer 2 Report

The authors have responded to the most of points.